# RNAi Analysis of Potential Functions of Cyclin B3 in Reproduction of Male Oriental River Prawns (*Macrobrachium nipponense*)

**DOI:** 10.3390/ani13101703

**Published:** 2023-05-21

**Authors:** Shubo Jin, Zhenyu Zhou, Wenyi Zhang, Yiwei Xiong, Hui Qiao, Yongsheng Gong, Yan Wu, Sufei Jiang, Hongtuo Fu

**Affiliations:** 1Wuxi Fisheries College, Nanjing Agricultural University, Wuxi 214081, China; 2Key Laboratory of Freshwater Fisheries and Germplasm Resources Utilization, Ministry of Agriculture and Rural Affairs, Freshwater Fisheries Research Center, Chinese Academy of Fishery Sciences, Wuxi 214081, China; 3Agriculture and Rural Bureau of Hanjiang District, Yangzhou 225007, China

**Keywords:** *Macrobrachium nipponense*, Cyclin B3, *IAG*, testis development, crustacean species

## Abstract

**Simple Summary:**

The rapid gonad reproduction of hatchlings restricts the sustainable development of *Macrobrachium nipponense*. Thus, it is urgently required to establish an artificial technique to regulate the process of gonad reproduction in *M. nipponense*. A previous study predicted that Cyclin B3 *(CycB3*) may perform crucial functions in the regulation of male reproduction in *M. nipponense*. In the present study, we aimed to investigate the potential roles of CycB3 in the male reproduction of this species. qPCR analysis results suggested that CycB3 was involved in the process of spermiogenesis, oogenesis, and embryogenesis in *M. nipponense*. RNA interference analysis showed that CycB3 affected the expression of insulin-like androgenic gland hormone and inhibited testis reproduction in *M. nipponense*. Taken together, these findings suggest that CycB3 plays essential roles in the regulation of male reproduction in *M. nipponense*, promoting the studies of the regulation of testis reproduction.

**Abstract:**

Cyclin B3 (*CycB3*) is involved in the metabolic pathway of the cell cycle, playing essential roles in the regulation of cell proliferation and mitosis. *CycB3* is also predicted to be involved in the reproduction of male oriental river prawns (*Macrobrachium nipponense*). In this study, the potential functions of *CycB3* in *M. nipponense* were investigated using quantitative real-time PCR, RNA interference, and histological observations. The full-length DNA sequence of *CycB3* in *M. nipponense* was 2147 base pairs (bp) long. An open reading frame of 1500 bp was found, encoding 499 amino acids. A highly conserved destruction box and two conserved cyclin motifs were found in the protein sequence of *Mn-CycB3*. Phylogenetic tree analysis revealed that this protein sequence was evolutionarily close to that of *CycB3*s of crustacean species. Quantitative real-time PCR analysis results suggested that *CycB3* was involved in the process of spermiogenesis, oogenesis, and embryogenesis in *M. nipponense*. RNA interference analysis showed that *CycB3* had a positive regulatory relationship with insulin-like androgenic gland hormone (*IAG*) in *M. nipponense*. In addition, sperm were rarely observed in the testis of *double-stranded CycB3*-injected prawns after 14 days of treatment, and sperm abundance was dramatically lower than that in the *double-stranded GFP*-injected prawns on the same day. This result indicated that *CycB3* can regulate the testis reproduction in *M. nipponense* through inhibiting the *IAG* expressions. Overall, these results indicated that *CycB3* plays essential roles in the regulation of male reproduction in *M. nipponense*, which may promote the studies of male reproduction in other crustacean species.

## 1. Introduction

The oriental river prawn (*Macrobrachium nipponense*) is a commercially important freshwater prawn in China that generates huge economic benefits [1]. It is widely distributed in freshwater and low-salinity estuarine regions. The annual aquaculture production of *M. nipponense* reached 225,321 tons in 2019, and the main aquaculture regions are Jiangsu, Anhui, and Zhejiang provinces [2]. Both the testis and ovary of *M. nipponense* reach sexual maturity within 40 days after hatching [3]. The rapid gonad development of hatchlings causes mating and propagation of multiple generations in the same ponds, resulting in prawns with smaller market size [4,5]. Therefore, an artificial technique to regulate the process of male reproduction is urgently needed to maintain the sustainable development of the *M. nipponense* aquaculture industry.

The eyestalk–androgenic gland–testis endocrine axis is involved in the regulation of gender differentiation and reproduction in male crustaceans [6,7]. The X-organ-sinus gland complex was found in the eyestalk of many crustaceans. It is considered as a principal neuroendocrine complex which can store and release many neurosecretory hormones [8]. These hormones play essential roles in the regulation of reproduction in crustaceans [9,10,11]. In *M. nipponense*, Jin [12] and Qiao [13] showed that crustacean hyperglycaemic hormone and gonad-inhibiting hormone have regulatory effects on gonad reproduction in *M. nipponense*. 

In male *M. nipponense*, the ablation of eyestalks stimulates the expression of insulin-like androgenic gland hormone (*IAG*) [4] and promotes testis development [5]. Meanwhile, the genes have been proven to positively regulate male reproduction in this species, of which the expressions were stimulated after the eyestalk ablations [14,15,16]. Thus, genes up-regulated after eyestalk ablation may affect male reproduction. Previous analysis showed cell cycle as the main metabolic pathway of differentially expressed genes after eyestalk ablation, suggesting its role in male reproduction of *M. nipponense* [4]. Cyclin B3 (*CycB3*) was a significantly up-regulated gene after eyestalk ablation, which was enriched in cell cycle, suggesting the potential role of *CycB3* in the promotion of male reproduction in this species. 

The process of gametogenesis plays a crucial role in the development of gonads in multicellular organisms. Several cell-cycle regulators were identified to regulate the process of gametogenesis, including cyclins (*Cycs*), cyclin-dependent kinases (*CDKs*), and cyclin-dependent kinase inhibitors. Cyclins play vital roles in cell proliferation in eukaryotic organisms by regulating the expression of *CDKs* [17]. In eukaryotic cells, the maturation promotion factor (MPF) can stimulate both mitotic and meiotic cell cycles. Thus, it is considered as a key regulator to affect cell proliferation [18]. MPF is a heterodimer composed of *CycB* and *CDK1* [19,20,21]. *CycB* is required during cell proliferation, which can activate or inhibit the activities of MPF [22]. This process is strongly related to the cell cycle and is important for *CDK1* activation. Three CycB isoforms have been reported in animals (*CycB1*, *CycB2*, and *CycB3*). *CycB3* is a mitotic cyclin that shares homology with A- and B-type cyclins. *CycB3* was reported to be associated with *CDK1* in chickens and fruit flies (*Drosophila*) [23,24], and it was shown to be involved in both oogenesis [25,26] and spermatogenesis [27].

In this study, quantitative real-time PCR (qPCR), in situ hybridization, RNA interference (RNAi), and histological observations were used to analyse the potential functions of *CycB3* in male reproduction of *M*. *nipponense*. The results of this study highlighted the functions of *CycB3* in *M*. *nipponense* and provided a basis for further studies of the mechanisms involved in male reproduction in other crustacean species.

## 2. Methods and Materials

### 2.1. Ethics Statement

We were permitted by the Institutional Animal Care and Use Ethics Committee of the Freshwater Fisheries Research Center, Chinese Academy of Fishery Sciences (Wuxi, China) to conduct experiments involving *M. nipponense* (Authorization NO. 20210715004, 15 July 2021). Dapu *M. nipponense* Breeding Base in Wuxi, China (120°13′44″ E, 31°28′22″ N) provided the prawns during both the reproductive season and non-reproductive season. The non-reproductive season was identified as January, with a water temperature of 13 ± 2 °C and illumination time of ≤12 h, while July was identified as the reproductive season with a water temperature of 30 ± 2 °C and illumination time of ≥16 h. Prior to tissue collection, prawns were maintained in aerated freshwater for 3 days with dissolved oxygen content ≥ 6 mg/L. Tissues were collected after prawns were anesthetized using an ice bath (approximate 2 °C).

### 2.2. Rapid Amplification of cDNA ends (RACE)

Testis were collected from male *M. nipponense* to synthesize the template for 3′ cDNA and 5′ cDNA cloning. Previous studies have described the detailed procedures for RACE cloning [28,29]. Briefly, total RNA was extracted from the testis using RNAiso Plus Reagent (Takara Bio Inc., Shiga, Japan). The 3′-Full RACE Core Set Ver.2.0 Kit and the 5′-Full RACE Kit (Takara) were used to synthesize the templates for 3′ cDNA and 5′ cDNA cloning using the extracted total testis RNA. The primers used for *Mn-CycB3* cloning (Table 1) were designed via the Primer-BLAST tool in NCBI (http://www.ncbi.nlm.nih.gov/tools/primer-blast/, accessed on 9 November 2021). Verification of the full-length cDNA sequence was conducted using two primer pairs (Table 1). ComputepI/Mwtool (http://ca.expasy.org/tools/pi_tool.html, accessed on 13 November 2021) was used to measure the theoretical isoelectric point and molecular weight of *Mn-CycB3* protein. The structural characteristics of *Mn-CycB3* were analysed with Blastx and Blastn (http://www.ncbi.nlm.nih.gov/BLAST/, accessed on 15 November 2021) and ORF Finder tool (http://www.ncbi.nlm.nih.gov/gorf/gorf.html, accessed on 15 November 2021). Table 2 provides accession numbers of amino acid sequences from different species used for the construction of the phylogenetic tree. MEGA X was utilized to construct the tree, after which the maximum-likelihood method and 1000 bootstrap replications were applied. 

### 2.3. The qPCR Analysis

The qPCR was performed in the different mature tissues and developmental stages, as well as in the testis and androgenic glands sampled during both the reproductive and non-reproductive seasons, in order to measure the relative mRNA expressions of *Mn-CycB3*. Fifty male *M. nipponense* (body weight of 3.45–4.32 g) and fifty female *M*. *nipponense* (body weight of 2.54–3.37 g) were used for this analysis. Eyestalks, brains, hearts, hepatopancreas, muscle, gonads, and gills were collected from both male and female prawns. Specimens at different developmental stages were collected from the full-sib population every 5 days during their maturation process. The testis and androgenic glands were collected during both the non-reproductive season and the reproductive season. Tissue samples were collected and pooled together (N = 5), in order to form a biological replicate. Six biological replicates were performed for qPCR analysis. Liquid nitrogen was used to preserve the collected tissues for qPCR analysis.

Previous studies have described the detailed procedures of RNA isolation and cDNA synthesis [28,29]. Briefly, according to the manufacturer’s protocol, the PrimeScript™ RT Reagent Kit (Takara) was employed to synthesize the cDNA template after the total RNA was extracted from each tissue with a UNlQ-10 Column Trizol Total RNA Isolation Kit (Sangon, Shanghai, China), which was then used to determine the expression level by applying the UltraSYBR Mixture (CWBIO, Beijing, China). The Bio-Rad iCycler iQ5 Real-Time PCR System (Bio-Rad, Hercules, CA, USA) was employed to perform the qPCR analyses in the present study. All qPCR reactions were run using three technical replicates. The thermal profile for qPCR was 95 °C for 10 min, followed by 40 cycles of 95 °C for 15 s and 60 °C for 1 min. DEPC-treated water was used to instead the template as a negative control. All primers used for the PCR analysis were listed in Table 1, including the Eukaryotic translation initiation factor 5A (*EIF*), which was used to normalize the transcript level of the target gene [30]. The amplification efficiencies of the target gene and reference gene were measured, and they were almost the same. The relative mRNA expressions of *Mn-CycB3* were calculated using the 2^−∆∆CT^ comparative CT method [31].

### 2.4. RNAi Analysis

RNAi was used to investigate the potential functions of *Mn-CycB3* in male *M. nipponense* reproduction. Specific RNAi primers were designed with a T7 promoter site using Snap Dragon (http://www.flyrnai.org/cgibin/RNAifind_primers.pl accessed on 18 June 2022), and synthesized into *Mn-CycB3* double-stranded RNA (*dsCycB3*) and *GFP* dsRNA (*dsGFP*) (negative control) [32] by using the Transcript Aid™ T7 High Yield Transcription kit (Fermentas, Inc., Waltham, MA, USA). 

Six hundred male *M*. *nipponense* were collected and randomly divided into two groups. One group was the *dsCycB3* group (RNAi), and the other group was the *dsGFP* group (control) (N = 300). These male prawns were collected at approximately 5 months after hatching and had a body weight of 3.48–4.56 g. The injected dose of *dsCycB3* and *dsGFP* was 4 μg/g according to the description in previous studies [33,34]. The same dose of each was injected into prawns 7 days after the first injection. Androgenic gland samples were collected from prawns in both groups on days 1, 7, and 14 after the first injection. The procedures for tissue collection and qPCR analysis are described above in Section 2.3. Both the *Mn-CycB3* and *Mn-IAG* mRNA expression levels were measured by qPCR to analyse the regulatory relationship between *CycB3* and *IAG* in *M. nipponense*.

### 2.5. Histological Observations

The morphological differences in the testis taken from *dsCycB3*- and *dsGFP*-injected prawns were measured by histological observation of tissues stained with hematoxylin and eosin (HE). The tissues were fixed in 4% paraformaldehyde prior to histological observations. Previous studies have described the detailed procedures of HE staining [35,36]. Briefly, tissues were dehydrated in varying ethanol concentrations, embedded in paraffin, and sliced to 5 µm thickness using a slicer (Leica, Wetzlar, Germany). The resulting sections were stained with HE for 3–8 min and viewed under an Olympus SZX16 microscope (Olympus Corporation, Tokyo, Japan).

### 2.6. Statistical Analysis

Data analysis was performed using SPSS Statistics 23.0 (IBM, Armonk, NY, USA). The independent *t*-test was used to compare data from control and RNAi groups on the same day. Statistical differences were determined by analysis of variance, followed by least significant difference and Duncan’s multiple range tests. Quantitative data were presented as mean ± standard deviation, of which *p*-values < 0.05 were considered statistically significant.

## 3. Results

### 3.1. Sequence Analysis

The full-length DNA sequence of *Mn-CycB3* was 2147 base pairs (bp) long, with a 5′ untranslated region of 114 bp and a 3′ untranslated region of 533 bp. The ORF was 1500 bp long and encoded 499 amino acids (Figure 1). The *Mn-CycB3* sequence was submitted to NCBI with the accession number OP379747.1. The theoretical isoelectric point and molecular weight of Mn-CycB3 were 9.07 and 57.066 kDa, respectively. The Blastx analysis in NCBI revealed that the protein sequence of *Mn-CycB3* shared over 65% identity with the *CycB3* protein sequence from other crustacean species, including *Penaeus monodon* (66.86%), *Penaeus japonicus* (66.53%), *Penaeus chinensis* (66.40%), and *Procambarus clarkia* (65.54%). A highly conserved destruction box was found at aa 73-81. Additionally, two conserved cyclin motifs were found at aa 285–369 and aa 382–463, respectively (Figure 2). 

### 3.2. Phylogenetic Tree Analysis

Ten well-defined protein sequences of *CycB3* from other aquatic animals were identified in NCBI using Blastx analysis (Table 2). The evolutionary distance between *Mn-CycB3* and the other species was analysed by constructing a condensed phylogenetic tree based on the protein sequences of these *CycB3*s. The phylogenetic tree contained two main branches consisting crustacean species on one and insect species on the other. The *Mn-CycB3* protein sequence clustered in the crustacean branch, and it had the closest evolutionary distance with those of penaeid shrimp species (Figure 3).

### 3.3. The qPCR Analysis

The qPCR analysis revealed that the *Mn-CycB3* mRNA expressions were detected in all tested tissues in the present study, indicating that CycB3 has multiple biological functions in *M. nipponense*. The *Mn-CycB3* mRNA was the highest in the testis of male prawns and ovary of female prawns, and the significant difference was observed between the testis and ovaries with the other tested tissues (*p* < 0.01). The expressions in the testis and ovary were 202.25-fold and 899.95-fold higher than that found in male muscle tissue, respectively, which had the lowest expression of all of the tested tissues. The *Mn-CycB3* mRNA showed higher expressions in the heart and hepatopancreas of male prawns than those of female prawns, while the opposite expression patterns were observed in the muscle, gonads, and gills (*p* < 0.01). The expression in the eyestalk and brain did not differ significantly between the sexes (*p* > 0.05) (Figure 4A). 

Extremely high expression of *Mn-CycB3* mRNA was observed at the cleavage stage during embryonic development, and the level differed significantly from the other tested stages (*p* < 0.01). No significant differences were detected among the other tested stages (*p* > 0.05). The *Mn-CycB3* mRNA expression level at the cleavage stage was 216.25-fold higher than that at the post-larval 25 stage, which had the lowest expression during the whole developmental process of juvenile prawns. A generally higher expressions of *Mn-CycB3* mRNA were detected during the embryonic developmental stages, compared to those of the larval and post-larval developmental stages (Figure 4B).

The expression levels of *Mn-CycB3* mRNA were also determined in the testis and androgenic gland between the reproductive season vs. non-reproductive season. The qPCR analysis showed that the expressions of *Mn-CycB3* mRNA in the testis and androgenic gland were 4.12-fold and 2.98-folder higher, respectively, during the reproductive season than during the non-reproductive season (*p* < 0.01) (Figure 5).

### 3.4. RNAi Analysis

The qPCR analysis revealed that *Mn-CycB3* remained at a stable level in the *dsGFP*-injected prawns and did not differ significantly over time (*p* > 0.05). However, the *Mn-CycB3* expression levels decreased significantly in the *dsCycB3*-injected prawns at days 7 and 14. The decrease reached 90% compared to the level in *dsGFP*-injected prawns on the same day (*p* < 0.01) (Figure 6A). The qPCR analysis also showed that the *Mn-IAG* expression level decreased with the decrease of *Mn-CycB3*. The decrease reached > 55% in the *dsCycB3*-injected prawns at days 7 and 14 compared to the level in *dsGFP*-injected prawns on the same day (*p* < 0.01) (Figure 6B).

HE staining revealed morphological differences in the testis between the *dsCycB3*- and *dsGFP*-injected prawns. According to the histological observations, three cell types were observed in the testis, including spermatogonium, spermatocyte, and sperm. The shape of spermatogonium is round. The shape of spermatocyte is also round, while it is slightly smaller than that of spermatogonium. The characteristics of sperm are non-flagellar and funnel-shaped sperm. Sperm contained a cone-shaped head part and a spiny part. The cell types in the testis of *dsGFP*-injected prawns did not differ over time. Sperm were the dominant cells and their abundance was dramatically higher than that of spermatogonia and spermatocytes (Figure 7). In the *dsCycB3*-injected prawns, the number of sperm decreased over time, and sperm were rarely observed at day 14, while spermatogonia and spermatocytes were the dominant cell types during this period (Figure 7).

*CycB3* is involved in the metabolic pathway of the cell cycle, and it plays essential roles during mitosis [37]. It is also involved in the regulation of both oogenesis [25,26] and spermatogenesis [27]. In a previous study, *CycB3* expressions were observed to be significantly up-regulated after the ablation of eyestalk from male *M. nipponense*, and thus *CycB3* was predicted to regulate the male reproduction of *M. nipponense* [4]. In the present study, we further investigated the potential regulatory roles of *CycB3* in the reproduction of male *M. nipponense*.

The Blastx analysis identified over 65% identity between the protein sequence of *Mn-CycB3* and the other well-defined *CycB3* protein sequences from the other species in NCBI. In addition, some typically conserved domains of *CycB3* were observed in the protein sequence of *Mn-CycB3*, including a highly conserved destruction box and two conserved cyclin motifs. This evidence indicated that the correct *Mn-CycB3* sequence was obtained. According to the phylogenetic tree analysis, the protein sequence of *Mn-CycB3* was closely related to those of other crustacean species, whereas the evolutionary distance from insect species was dramatically long. More *CycB3* sequences from freshwater prawns should be investigated to improve the evolutionary analysis of *CycB3*.

In humans, *CycB3* mRNA was detected in all tested tissues but was significantly abundant in the testis [38]. *CycB3* mRNA expression was also reported in some aquatic animals. For example, *CycB3* mRNA expression was highest in the gonad of the Pacific oyster (*Crassostrea gigas*) and it increased with gonad development, indicating that *CycB3* was involved in the process of oogenesis and spermatogenesis in this species [26]. *CycB3* was also dominantly expressed during spermatogenesis in the Japanese eel (*Anguilla japonica*) [39]. In the present study, *Mn-CycB3* mRNA expression was significantly higher in the testis and ovary of male and female prawns, respectively, compared to the levels in the other tested tissues. This result suggests that *CycB3* played essential roles in the process of oogenesis and spermatogenesis in *M. nipponense*. Furthermore, the *Mn-CycB3* mRNA showed higher expression in the testis and androgenic gland taken from the reproductive season than those from the non-reproductive season. Previous studies identified the significant morphological differences in the testis and androgenic gland between the two seasons, with more vigorous tissue development during the reproductive season [40,41]. This evidence confirmed that *CycB3* was involved in the process of spermatogenesis in *M. nipponense*, which is consistent with reports about other species [26,39]. 

*Mn-CycB3* mRNA was detected during the whole developmental process of juvenile prawns, indicating that *CycB3* had multiple functions in the promotion of *M. nipponense* development. However, its expression was generally higher during the embryonic developmental stages than during larval and PL development, which supported the premise that *CycB3* regulated the process of embryogenesis in *M. nipponense* [14,15,16]. Additionally, its expression peaked at the cleavage stage, suggesting that cell proliferation (mitosis) was extremely vigorous during this period.

Knockdown of the expression of *CycB3* by RNAi inhibited the process of ovarian development in the silk moth *Bombyx mori* [25]. However, to the best of our knowledge, RNAi analysis of the functions of *CycB3* in male reproduction has not been reported previously for all species. RNAi has been widely used to analyse gene functions in *M. nipponense*, including male reproduction-related genes [42,43,44]. In the present study, qPCR analysis revealed that *dsCycB3* injection resulted in significant decreases of *Mn-CycB3* expression at days 7 and 14, indicating that synthesized *dsCycB3* can efficiently knockdown the expression of *CycB3* in *M. nipponense*. The decreased *Mn-CycB3* expression also led to decreased *Mn-IAG* expression, indicating that *CycB3* positively regulated *IAG* expression in *M. nipponense*. Androgenic gland is a special organ, existed in male crustaceans. The androgenic gland and its secreted hormones have been proven to play essential roles in the regulation of male differentiation and reproduction of crustaceans, especially the formation of testis and the secondary male sexual characteristics [45,46,47]. *IAG* is the main expressed gene in the androgenic gland, which was reported to have positive regulatory roles on male differentiation and development in crustacean species [47,48]. The functions of *IAG* have been well-identified in crustaceans such as *Fenneropenaeus chinensis* [49], *Scylla paramamosain* [50], *Lysmata vittata* [51], *Fenneropenaeus merguiensis* [52], and *M. nipponense* [53]. Knockdown *IAG* expression by RNAi produced a marked inhibitory effect on the process of spermatogenesis in *Macrobrachium rosenbergii* [54]. Thus, the positive relationship between *CycB3* and *IAG* suggests that *CycB3* has potentially regulatory effects on the reproduction of male *M. nipponense*. The significant morphological differences were observed in the testis between *dsGFP-* and *dsCycB3*-injected prawns, revealed by the histological observations. Sperm were rarely observed at day 14 after *dsCycB3* injection, whereas sperm were the dominant cells in the *dsGFP-*injected prawns on the same day. This result indicated that knockdown of the expression of *CycB3* inhibited testis development in *M. nipponense*. Overall, our data show that *CycB3* regulated the testis reproduction by affecting *IAG* expression in *M. nipponense*.

## 4. Conclusions

In conclusion, the present results highlighted the important roles of *CycB3* to regulate the process of reproduction in male *M. nipponense*, as verified by qPCR analysis, RNAi, and histological observations. *CycB3* showed the highest expressions in the testis of male prawns and ovaries of female prawns. In addition, higher expressions of *CycB3* were observed in the testis and androgenic gland taken from the reproductive season, compared to those of the non-reproductive season. The above results suggested that *CycB3* may regulate the gonad reproduction in *M. nipponense*. RNAi analysis revealed that knockdown of the expressions of *CycB3* also leads to the decrease of *IAG*, and sperm were rarely found at Day 14 after the injection of *dsCycB3*, which were dramatically lower than those of the *dsGFP*-injected group on the same day, indicating that *CycB3* regulates testis development through inhibiting the *IAG* expression in this species. This study provided valuable data that can be applied to establish an artificial technique for regulating testis development in *M. nipponense*.

## Figures and Tables

**Figure 1 animals-13-01703-f001:**
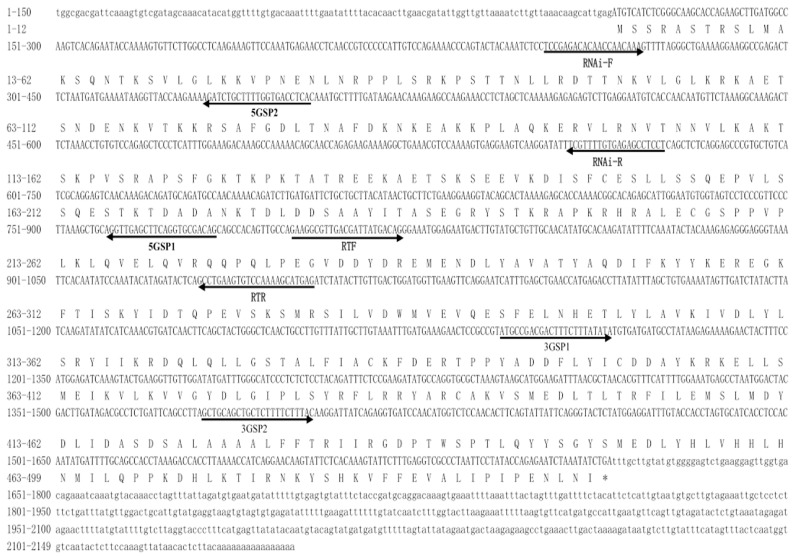
Nucleotide and deduced amino acid sequence of *Mn-CycB3*. Both the nucleotide and deduced amino acid sequence are displayed in the 5′–3′ directions. Lowercase letters indicated the 3′ UTR and 5′ UTR, while the open reading frames are shown in capital letters. A single capital letter indicated the amino acid code of the deduced amino acid sequence. The Methionine (ATG) denoted the initiation codon, and the termination codon (TGA) was shown as an asterisk.

**Figure 2 animals-13-01703-f002:**
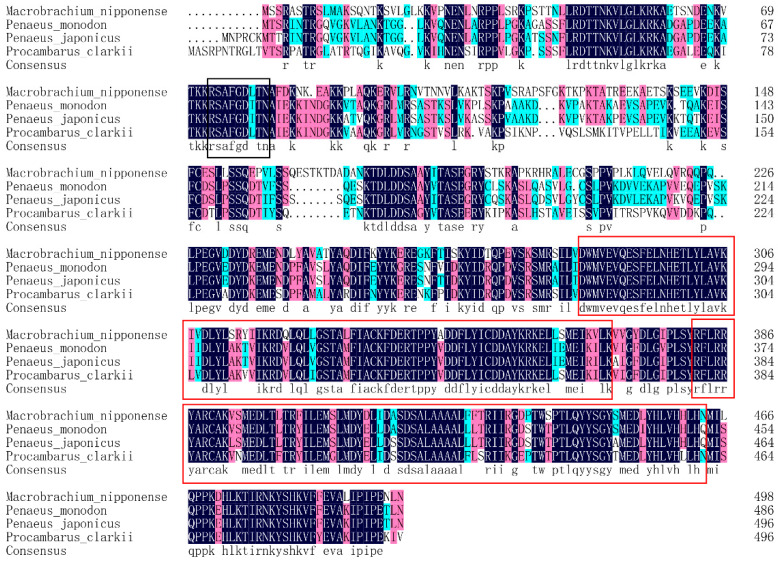
The similarity identity of amino acid sequences of *CycB3* between different species. Black boxes indicate the conserved destruction box. Red boxes indicated the conserved cyclin motifs. The alphabets with black indicate that the amino acids between different species are the same, while the alphabets with the other colours indicate that the amino acids between different species are different.

**Figure 3 animals-13-01703-f003:**
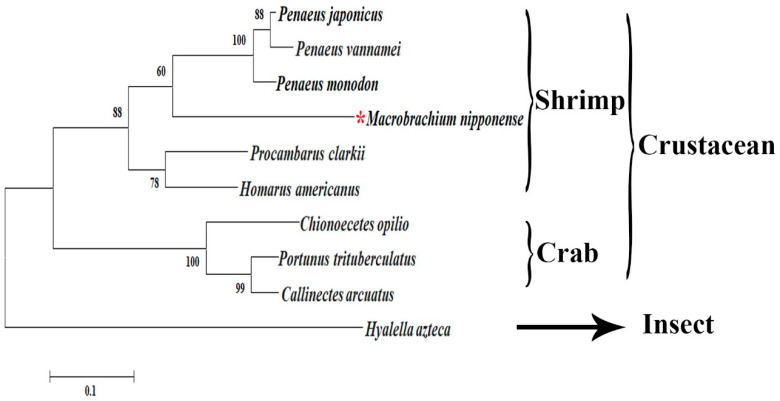
Phylogenetic tree of amino acid sequences of *CycB3* from various species. *M. nipponense* was marked by red asterisk.

**Figure 4 animals-13-01703-f004:**
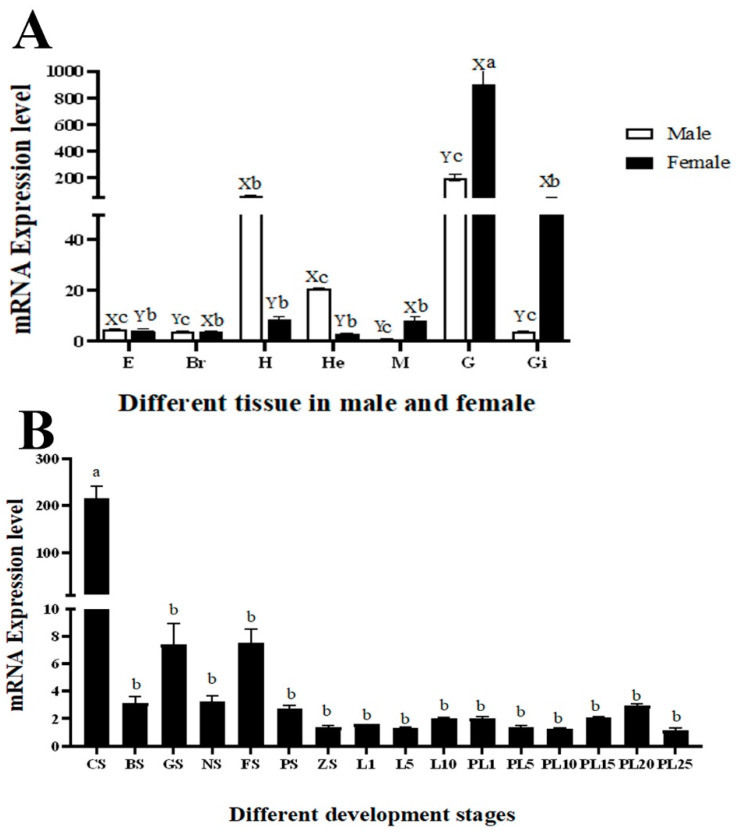
Expression analysis of the *Mn-CycB3* in different mature tissues (**A**) and developmental stages (**B**) of *M. nipponense* by qPCR. The *EIF* was used to normalize the amount of *Mn-CycB3* transcript level. Data are expressed as the mean ± SD (n = 6). Lowercases were used to indicate differences in *Mn-CycB3* expression between different samples, while uppercase letters were used to indicate differences in *Mn-CycB3* expression between male and female prawns within the same tissue. E, Br, H, He, M, G, and Gi indicate eyestalk, brain, heart, hepatopancreas, muscle, gonad and gill, respectively. CS, BS, GS, NS, FS, PS, GS, L, and PL denote cleavage stage, blastula stage, gastrula stage, nauplius stage, posterior nauplius stage, protozoea stage, zoea stage, larval developmental stage, and post-larval developmental stage, respectively.

**Figure 5 animals-13-01703-f005:**
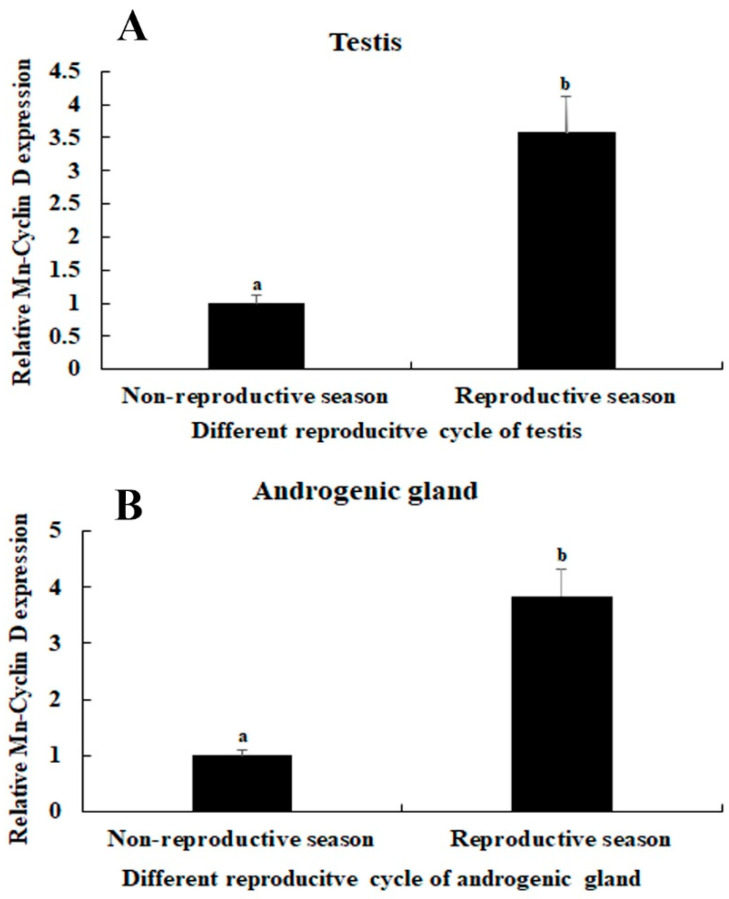
Expression analysis of the *Mn-CycB3* in the testis (**A**) and androgenic gland (**B**) of *M. nipponense* taken from different reproductive season. The *EIF* was used to normalize the amount of *Mn-CycB3* transcript level. Data are expressed as the mean ± SD (n = 6). Lowercases are used to indicate differences in *Mn-CycB3* expression between different samples.

**Figure 6 animals-13-01703-f006:**
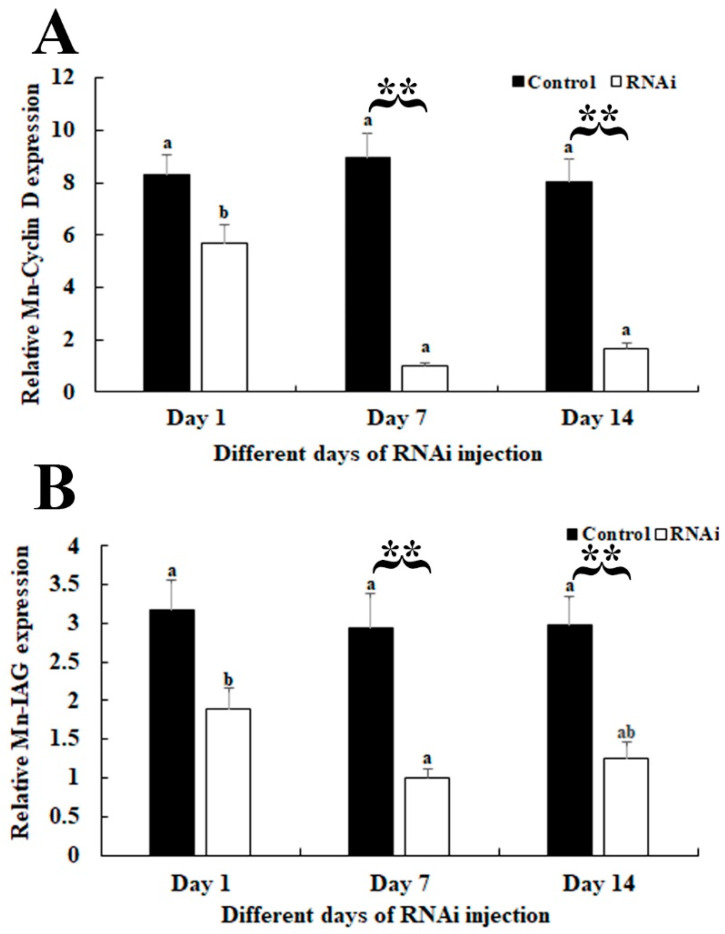
Expression analysis of *Mn-CycB3* (**A**) and *Mn-IAG* (**B**) of *M. nipponense* after the injection of *dsCycB3* and *dsGFP*. The *EIF* was used to normalize the amount of *Mn-CycB3* transcript level. Data are expressed as the mean ± SD (n = 6). Lowercases are used to indicate differences in gene expression between different days after the injection of *dsGFP* and *dsCycB3*. ** (*p* < 0.01) is used to indicate significant differences in *Mn-CycB3* and *Mn-IAG* expression between the RNAi group and control group on the sample day.

**Figure 7 animals-13-01703-f007:**
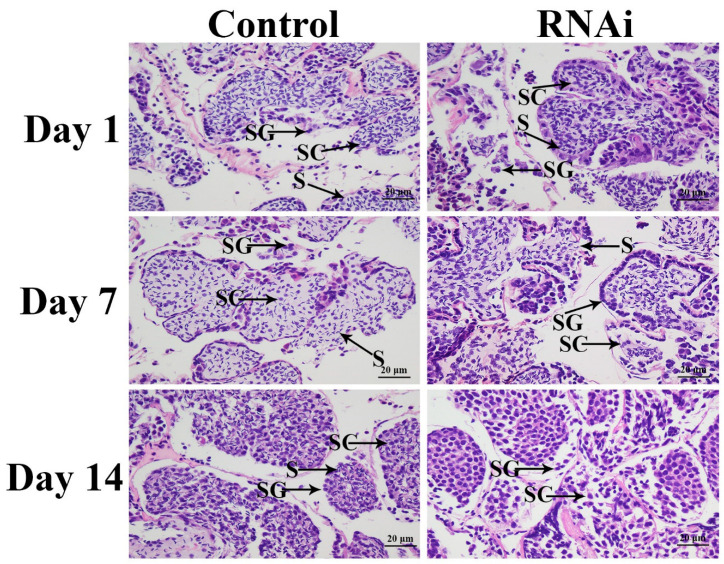
The histological observations of testis of *M. nipponense* between *dsGFP*-injected and *dsCycB3*-injected prawns. SG, SC, and S denote spermatogonium, spermatocyte, and sperm, respectively. Scale bars = 20 μm.

**Table 1 animals-13-01703-t001:** Universal and specific primers used for PCR amplification and qPCR analysis.

Primer Name	Nucleotide Sequence (5′→3′)	Purpose
CycB3-3GSP1	ATGCCGACGACTTTCTTTATAT	FWD first primer for CycB3 3′ RACE
CycB3-3GSP2	GCTGCAGCTGCTCTTTTCTTTA	FWD second primer for CycB3 3′ RACE
CycB3-5GSP1	CTGTCGCACCTGAAGCTCAACC	RVS first primer for CycB3 5′ RACE
CycB3-5GSP2	GTGAGGTCACCAAAAGCAGATC	RVS second primer for CycB3 5′ RACE
3′RACE OUT	TACCGTCGTTCCACTAGTGATTT	RVS first primer for 3′ RACE
3′RACE IN	CGCGGATCCTCCACTAGTGATTTCACTATAGG	RVS second primer for 3′ RACE
5′RACE OUT	CATGGCTACATGCTGACAGCCTA	FWD first primer for 5′ RACE
5′RACE IN	CGCGGATCCACAGCCTACTGATGATCAGTCGATG	FWD second primer for 5′ RACE
CycB3-RTF	GAAGGCGTTGACGATTATGACAG	FWD primer for CycB3 expression
CycB3-RTR	CTCATGCTTTTGGACACTTCAGG	RVS primer for CycB3 expression
IAG-RTF	CTGACCACACCTACTGAAGACAA	FWD primer for IAG expression
IAG-RTR	CGTTTTCGATAAGAGGTCAAGCC	RVS primer for IAG expression
EIF-F	CATGGATGTACCTGTGGTGAAAC	FWD primer for EIF expression
EIF-R	CTGTCAGCAGAAGGTCCTCATTA	RVS primer for EIF expression
CycB3 RNAi-F	TAATACGACTCACTATAGGGTCCGAGACACAACCAACAAA	FWD primer for RNAi analysis
CycB3 RNAi-R	TAATACGACTCACTATAGGGAGGAGGCTCTCACAAAACGA	RVS primer for RNAi analysis

**Table 2 animals-13-01703-t002:** Sequences used for phylogenetic tree analysis.

Species	Accession Number
*Macrobrachium nipponense*	
*Penaeus monodon*	XP_037786045.1
*Penaeus japonicus*	XP_042878469.1
*Procambarus clarkii*	XP_045600532.1
*Homarus americanus*	XP_042217784.1
*Penaeus vannamei*	XP_027238877.1
*Portunus trituberculatus*	XP_045137868.1
*Callinectes arcuatus*	QPO25106.1
*Chionoecetes opilio*	KAG0693500.1
*Hyalella azteca*	XP_018006502.1

## Data Availability

The data generated and analyzed during this study are included in this article.

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
