# Peer review of "RNAi Analysis of Potential Functions of Cyclin B3 in Reproduction of Male Oriental River Prawns (*Macrobrachium nipponense*)"

_animals, 2023, doi:10.3390/ani13101703_

Round 1
Reviewer 1 Report
In my opinion, the manuscript is of very high quality. However, it requires a minor revision before approval. The most serious remark concerns the lack of Simple summary. The second - ways of presenting citations references. All my comments are in the text. To see them all, open the file in Acrobat Reader.

Author Response
Reviewer 1
In my opinion, the manuscript is of very high quality. However, it requires a minor revision before approval. The most serious remark concerns the lack of Simple summary. The second - ways of presenting citations references. All my comments are in the text. To see them all, open the file in Acrobat Reader.
I can't find Simple summary.
Reply: Thank you for your reminding. The simple summary with 154 words has been provided in the main text.
The references should be present as "normal" not upperscript - please correct in whole manuscript.
Reply: Thank you for your reminding. All of the references have been revised as normal.
2 °C?
Reply: Yes, the temperature for the ice bath is approximate 2°C. We have provided it at line 114 in the manuscript.
for ... please add more details
Reply: Thank you for your reminding. The table legend for Table 1 has been revised as “Table 1. Universal and specific primers used for PCR amplification and qPCR analysis”.
see remarks on Table 1
Reply: Thank you for your reminding. The table legend for Table 2 has been revised as “Table 2. Sequences used for phylogenetic tree analysis”.
add English and Latin name of the species
Reply: The Latin name of the species has been provided in the legend of figure 4-7.
Present
Reply: Thank you for your reminding. We have revised it in the manuscript.
Reviewer 2 Report
General comments
In this paper, the authors investigated the cloning of a cDNA encoding cyclin B3 (Mn-CycB3) from the oriental river prawn Macrobrachium nipponense. Mn-CycB3 was highly expressed in gonads such as the testis and ovary. Expression levels of Mn-CycB3 in the testis and androgenic gland during the reproductive season were higher than those during the non-reproductive season. Gene silencing of Mn-CycB3 suppressed the gene expression of insulin-like androgenic gland hormone (Mn-IAG) in the androgenic gland and reduced sperm cells in the testis. Therefore, the authors suggest that Mn-CycB3 may control spermatogenesis in the testis via regulation of IAG expression. There are, however, some serious issues described below, and therefore the article at this form might not be acceptable for the publication in Animals.
Specific comments
1) Amino acid position “aa 71-79” in line 184 is incorrect. The authors should correct it to "aa 73-81".
2) “It had the closest evolutionary distance with freshwater prawns” in lines 192-193 is strange. Mn-CycB3 was clustered with those of penaeid shrimp species in the molecular phylogenetic tree (Figure 3), but not with that of the red swamp crayfish.
3) Some of the amino acid sequences in figure 2 are overlapped and not visible. In addition, there are no explanations for the color of alphabets in the figure legend.
4) Expression levels of Mn-CycB3 of the testis in figures 4-6 are quite different. The authors should explain these phenomena in the text.
5) Crustacean sperm shows a very characteristic shape. The authors should explain the morphology of the sperm in the testis from M. nipponense using the pictures in figure 7. Characteristics of spermatocyte and spermatogonia should also be explained.
6) Although the authors mentioned that the number of sperm in the testis reduced by Mn-CycB3 RNAi, no differences between control and RNAi were observed at 14 days after the injection (Figure 7).
Author Response
Reviewer 2
In this paper, the authors investigated the cloning of a cDNA encoding cyclin B3 (Mn-CycB3) from the oriental river prawn Macrobrachium nipponense. Mn-CycB3 was highly expressed in gonads such as the testis and ovary. Expression levels of Mn-CycB3 in the testis and androgenic gland during the reproductive season were higher than those during the non-reproductive season. Gene silencing of Mn-CycB3 suppressed the gene expression of insulin-like androgenic gland hormone (Mn-IAG) in the androgenic gland and reduced sperm cells in the testis. Therefore, the authors suggest that Mn-CycB3 may control spermatogenesis in the testis via regulation of IAG expression. There are, however, some serious issues described below, and therefore the article at this form might not be acceptable for the publication in Animals.
Specific comments
1) Amino acid position “aa 71-79” in line 184 is incorrect. The authors should correct it to "aa 73-81".
Reply: Thank you for your reminding. We have revised it in the manuscript at line 208.
2) “It had the closest evolutionary distance with freshwater prawns” in lines 192-193 is strange. Mn-CycB3 was clustered with those of penaeid shrimp species in the molecular phylogenetic tree (Figure 3), but not with that of the red swamp crayfish.
Reply: Thank you for your reminding. We have revised it as “The Mn-CycB3 protein sequence clustered in the crustacean branch, and it had the closest evolutionary distance with those of penaeid shrimp species” in the text at line 215-217.
3) Some of the amino acid sequences in figure 2 are overlapped and not visible. In addition, there are no explanations for the color of alphabets in the figure legend.
Reply: Thank you for your suggestion. We have revised Figure 2. The explanations for the alphabets with different colour were also provided in the legend of Figure 2. We hope it meets your demand now.
4) Expression levels of Mn-CycB3 of the testis in figures 4-6 are quite different. The authors should explain these phenomena in the text.
Reply: In the present study, the relative mRNA expressions of Mn-CycB3 were calculated using the 2−∆∆CT comparative CT method. This method is that the tissue with lowest expression was set as 1, and the other tissues were compared with the lowest tissue to calculate the expressions of the other tissues. Figure 4-6 measured the expressions of Mn-CycB3 under different conditions. Figure 4 measured the Mn-CycB3 expressions in different mature tissues and developmental stages. Figure 5 measured the Mn-CycB3 expressions in the testis and androgenic gland of reproductive season and non-reproductive season. Figure 6 measured the Mn-CycB3 and Mn-IAG expressions after the injection of dsGFP and dsCycB3. The possible explanations of qPCR analysis were also provided in the text of line 286-295.
5) Crustacean sperm shows a very characteristic shape. The authors should explain the morphology of the sperm in the testis from M. nipponense using the pictures in figure 7. Characteristics of spermatocyte and spermatogonia should also be explained.
Reply: Thank you for your reminding. The morphology of spermatocyte, spermatogonia and sperm were provided at line 253-257. We hope it is clear now.
6) Although the authors mentioned that the number of sperm in the testis reduced by Mn-CycB3 RNAi, no differences between control and RNAi were observed at 14 days after the injection (Figure 7).
Reply: According to the picture of “RNAi” at Day 14 in Figure 7, no sperm was observed at this period. According to the picture of “control” at Day 14 in Figure 7, at least 50% cells were observed as sperm. Sperm was shown as the arrow of “S”. Thus, the histological observations indicated the significant difference of cell types between the testis of control and RNAi at 14 days after the injection.
Reviewer 3 Report
This manuscript shows that the gene cloning and deduced protein phylogenetic tree analysis of Cyclin B, which is thought to be involved in male sexual maturation in M. nipponense, is homologous to the CycB3 gene in other crustaceans. Quantitative real-time PCR analysis suggested that the gene is involved in spermatogenesis, oogenesis, and embryogenesis. Furthermore, RNA interference analysis indicated that CycB3 affects the expression of IAG, a male sexual maturation hormone, and is involved in testicular development. This indicates that RNA interference of CycB3 used in this study is a useful analytical technique in clarifying the mechanism of male sexual maturation.
Author Response
This manuscript shows that the gene cloning and deduced protein phylogenetic tree analysis of Cyclin B, which is thought to be involved in male sexual maturation in M. nipponense, is homologous to the CycB3 gene in other crustaceans. Quantitative real-time PCR analysis suggested that the gene is involved in spermatogenesis, oogenesis, and embryogenesis. Furthermore, RNA interference analysis indicated that CycB3 affects the expression of IAG, a male sexual maturation hormone, and is involved in testicular development. This indicates that RNA interference of CycB3 used in this study is a useful analytical technique in clarifying the mechanism of male sexual maturation.
Reply: Thank you for your agreement.